# Supply chain integration and innovation performance of manufacturing firms: The moderating role of research and development investment intensity

**Juanmei Zhou, Jie Mei**◉*

School of Economics and Management, North University of China, Taiyuan, China

* 1152069909@qq.com

## Abstract

This paper delves into the impact of supply chain integration on corporate innovation performance. Utilizing panel data from 1,038 manufacturing companies listed on China's A-shares stock market from 2012 to 2021, it analyzes how firms influence their innovation performance through supply chain integration and explores the moderating role of R&D investment in this relationship. The study reveals that internal integration significantly enhances corporate innovation performance, while customer and supplier integration negatively impact it. Furthermore, R&D investment mitigates the negative effect of supplier integration on innovation performance and positively moderates the relationship between customer integration and innovation performance. Given the diversity in ownership structures and equity concentration among Chinese firms, heterogeneity analysis shows that the positive effect of internal integration on innovation performance is more pronounced in state-owned enterprises and firms with high equity concentration. Conversely, in non-state-owned enterprises and firms with low equity concentration, the negative impact of customer and supplier integration on innovation performance is more significant, and the moderating effect of R&D investment varies according to the firm's heterogeneity. The findings can help firms understand the mechanisms through which different dimensions of supply chain integration affect innovation. By leveraging resource dependence theory, this study offers theoretical guidance for supply chain management and offers practical insights aimed at enhancing corporate innovation practices.

## Introduction

At present, China's economic development model is changing from "factor-driven" to "innovation-driven" [1]. Innovation, as the primary driver of development, is conducive to maintaining competitive advantage and achieving sustainable development [2]. Due to the dual effects of technological progress, technological innovation actively promotes industrial innovation. The innovation ability of enterprises has become a crucial factor in their development and can even determine the fate of the country, especially with the current political backdrop and deepening economic globalization. According to the China Science and

**Data availability statement:** This paper used the DIB database and CSMAR database. All the data are primary data. CSMAR: https://data.csmar.com/ DIB database: https://www.dibdata.cn/d2/index.html.

**Funding:** This study was funded Humanities and Social Sciences Youth Foundation, Ministry of Education, (22YJC790142); Ministry of Education Humanities and social Science research project Youth Fund project, (23YJC790133); Key research base of Humanities and social Sciences in Shanxi Province, (20200128); Shanxi graduate excellent course project, (2023YZ31). The funders had no role in study design, data collection and analysis, decision to publish, or preparation of the manuscript.

**Competing interests:** The authors have declared that no competing interests exist.

Technology Funding Statistics Bulletin 2022, China's total R&D expenditure reached 3,087 billion yuan in 2022, which has been increasing for seven consecutive years. At the same time, enterprises, as the main body of R&D investment and transformation of scientific and technological products, contributed 76.9% to the overall increase in the national R&D expenditure, becoming the dominant force in ensuring the continued growth of national R&D funding. Although the amount of R&D investment by enterprises in China has been increasing, various phenomena still indicate that the overall innovation capacity of Chinese enterprises is weak, reflecting the lack of senior management and technical talent and the low output of innovative achievements. In response to this, the government and academics have been actively exploring the factors that affect business innovation as an important topic.

With the global information revolution and the growth of the international division of labor, the manufacturing industry's production patterns have dramatically shifted. Many products now result from cooperation across multiple companies in the supply chain, and competition has shifted from between individual enterprises to between supply chains [3]. Supply chain integration, as an important feature of the enterprise supply chain structure, emphasizes efficient collaboration between the enterprise and key suppliers and customers. Through strategic alliances and cooperative relationship with different partners in the supply chain, the enterprise can focus on its core competencies and complete its exclusive functions [4]. This strategy enables the efficient circulation of information flow, capital flow, and logistics within the overall supply chain, as well as the independent configuration of these flows. It is an innovative strategy in the development process of the enterprise [5]. Supply chain integration is becoming a new form of industrial management [6], impacting various aspects of business performance [7–12], organizational resilience [13], product quality [14], operational performance [15], and relationship stability [16]. Exploring the impact of supply chain integration on business activities, particularly from the perspective of enterprise innovation performance, has become an important topic in recent years. However, relevant research is still lacking, and existing studies have not yet reached a consistent conclusion.

In addition, the intensity of an enterprise's R&D investment also affects its innovation performance. In the process of innovation, enterprises need sufficient inputs to achieve corresponding output results. R&D inputs positively impact innovation through economies of scale and the accumulation of knowledge, technology and capital [17]. At present, a large number of studies by scholars focus on internal factors such as governance structure [18,19], property rights attributes [20], management style [21], and regulatory mechanism [22] of enterprises to explore the role mechanism between R&D investment and internal control of enterprises on innovation performance. Around the theoretical mechanism of supply chain affecting innovation, it has been found in existing studies that R&D investment plays an effective mediating role between supply chain and firms' innovation [23]. High investment in technological innovation does not guarantee high output, but empirical evidence suggests that firms' R&D investment can enhance innovation efficiency [24]. Some scholars also use R&D investment as a moderating variable to explore how it moderates the impact of environmental regulations [25], local officials' mobility [26], and knowledge-based network structure [27] on firms' innovation performance. However, in a highly competitive market environment, the influence of external factors posed by customers and suppliers should also be emphasized, and due to the growing business needs of companies in the supply chain, companies are facing new challenges in innovating how to transform the key resources acquired in the supply chain. Supply chain is an important factor affecting corporate innovation, except for Pan et al. [28], who found that supply chain efficiency positively affects innovation efficiency through the moderating effect of R&D investment, few studies have constructed a mechanism for the impact of R&D investment in the field of supply chain integration on innovation. This raises questions

for firms about whether the relationship between the integration of supply chain structure and firms' innovative activities is similarly moderated by this factor, given that the intensity of R&D investment has a significant impact on innovation. Therefore, this paper employs R&D investment as a moderating variable and supports this relationship with empirical evidence.

To address the research gap, this paper empirically examines the impact and mechanisms of supply chain integration on the innovation performance of firms, using data from 1038 listed companies in China's manufacturing industry. The study also explores the moderating effect of R&D investment intensity in the path of external integration through the mechanism test. Theoretically, resource dependence theory accommodates the simultaneous exploration of the firm's internal environment and external influences, not only explaining the importance of internal organizational coordination, but also emphasizing the interdependent collaboration between firms and their partners [29]. Therefore, using resource dependence theory as a theoretical foundation ensures that the integration of dimensions in the supply chain can be considered in a theoretical framework, and the impact of supply chain integration on firms' innovation performance can be fully understood. Furthermore, the coexistence of enterprises with different types of property rights is a major feature of Chinese enterprises. Whether an enterprise is state-owned or privately owned implies differences in market position and resource endowment. These differences, caused by the nature of property rights, influence the innovation activities of enterprises. Additionally, variations in equity concentration lead enterprise to adopt different management strategies. From the perspective of supply chain management, these different management decisions also affect the innovation performance of enterprises. The theoretical and practical contributions of this study achieve three objectives: First, from the perspective of enterprise supply chain structure, by constructing a panel vector autoregressive model, we empirically analyze the impact of supply chain integration, including internal integration, customer integration, and supplier integration dimensions on the innovation performance of enterprises. Few studies focus on the innovation effects among node firms within the supply chain, and this paper aims to reveal how the supply chain influences the innovation activities and the logical mechanism. Secondly, we clarify how R&D investment moderates the impact of customer integration and supplier integration on enterprise innovation performance. This understanding can help alleviate the pressures from suppliers and customers, solve management challenges in upstream and downstream relationships, and promote innovation within enterprises, thereby expanding research on supply chain management and enterprise innovation. Third, this study offers practical recommendations for firms with varying property ownership and equity concentrations in managing their supply chains. It enhances research pertaining to the impact of R&D investment intensity on firms' innovation and provides insights into enhancing internal governance, refining external integration strategies, and advancing innovation and development in manufacturing firms.

The structure of this paper is as follows: the second part starts from the theoretical foundation and conceptual division of this paper, provides a logical derivation of the arguments of this study, and formulates the research hypotheses. The third part describes the way of selecting each indicator and the related data sources, and sets up the model. The fourth part utilizes the multiple regression method to conduct the empirical analysis, and the empirical results are tested for robustness by replacing the core variable measures, endogeneity test, and panel-corrected standard error model. Part V summarizes the conclusions, theoretical contributions, practical contributions and limitations.

## Literature review and theoretical analysis

Most of the scholars consider supply chain integration as a holistic concept and believe that supply chain integration activities facilitate timely access to information about demand,

capabilities, and strategies, thus enhance the utilization of external relationships [30]. With further research, the process of supply chain integration has gradually evolved into a system optimization process, transforming from internalization to a comprehensive integration from inside to outside and from parts to the whole. From a supply chain management perspective, an enterprise can be viewed both as a whole consisting of different functional departments and as a node within the supply chain network. Therefore, the study of supply chain integration must include internal integration among functional departments, while clarifying the different roles of external integration with customers and suppliers in enterprise management. Only by subdividing the dimensions of supply chain integration and visualizing the abstract concept can we fully understand the mechanism of its relationship with innovation performance. To synthesize previous studies [31] and comprehensively consider supply chain integration, this paper discusses firms' innovation performance across three dimensions: internal integration, supplier integration and customer integration.

Past research on corporate innovation is generally rooted in endogenous growth theory, which argues that innovation performance improves with high factor inputs but overlooks the influence of external factors, such as the macroeconomic environment, government, and other organization. Therefore, this paper puts internal integration, customer integration and supplier integration under the same framework from Porter's industrial positioning theory and transaction cost theory under the framework of resource dependence theory to ensure a full understanding of the impact mechanism of the three dimensions of supply chain integration on corporate innovation. Resource dependence theory believes that the heterogeneous resources possessed by different enterprises are different, and the heterogeneous resources possessed are scarce and difficult to be imitated [32], and the innovation elements are acquired only in the exchange and cooperation between the enterprise and the outside world. Suppliers and customers in the supply chain provide complementary resources, making them important sources of innovation resources for the firm. However, the degree to which these resources can be obtained is influenced by the level of integration with these partners.

## Internal integration and firms' innovation performance

Resource Dependence Theory (RDT) posits that enterprise resources can be categorized into internal and external categories based on the organization's environment [33]. Internal resources encompass the enterprise's empirical knowledge and human capital. Effective integration of internal resources transforms new ideas into new products. It also ensures efficient access to and transformation of both internal and external knowledge and technological resources, facilitating innovations output and providing a key competitive advantage [34]. Moreover, the integration of internal resources facilitates synergy among functional departments, improving communication and optimizing the decision-making process [35]. Enterprise innovation can be categorized as exploratory innovation or utilization innovation. Utilization innovation, a form of technological innovation, involves passive adaptation to environmental changes and focuses on integration and utilizing existing internal knowledge and technology [36]. Whether measured by case collection method [37] or modeling analysis using validated factor analysis [38], it was equally found that internal integration in the supply chain integration process is more effective in driving corporate innovation. Manufacturing enterprises, due to their industrial structure, business models and organizational characteristics, exhibit interlocking operational traits. Internal integration allows manufacturing firms to closely connect R&D, production, sales and materials departments, using smooth information flow to eliminate internal barriers. This ensures that production feedback and market needs reach the R&D department directly, fostering innovation and technological advancement.

Unlike regular asset investments, innovation activities require larger investments and longer capital cycle, with higher risks and supervision costs. A sound internal integration mechanism can reduce these costs, providing resource support for innovation. Based on the above analysis, the following hypotheses are proposed.

H1 Internal integration has a positive correlation with firms' innovation performance.

## Customer integration and firms' innovation performance

Some scholars argue that customer integration positively impacts firms' innovation performance. Customers, being directly in front of consumer groups, are responsible for ensuring consumers are aware of and purchase the company's products. At the same time, enterprises can obtain feedback from customers on product design, usage performance, and other market elements, which not only enhances the visibility of external problems, but also helps enterprises gain a clear insight into market changes and needs, thus ensuring that enterprises can make better and faster decisions on supply chain management operations [39,40]. However, some scholars have overlooked the issue of fit between firms and their customers. As buyers, customers have the discretion to choose their collaborators, making it uncertain whether firms can obtain reliable and high-quality innovation resources from them. Skippari et al. [41] emphasizes the need for managers to focus on the mechanism of interaction between collaboration and innovation, and for decision makers to pay attention to the degree of fit between firms and collaborators during the social exchange process of collaborative innovation activities in supply chains to ensure that both parties are empowered to innovate through the sharing of information resources.

Therefore, this paper presents an opposing research conclusion, suggesting that customer integration in supply chain management impacts the firm's core profits, which subsequently negatively affects the firm's innovation performance. According to industrial organization theory and transaction cost theory, the main performance in the current supply chain network is that the buyer's power is more prominent, and the customer has more initiative in choosing the enterprise, and has a significant impact on enterprise pricing [42]. When customer integration is high, several key customers leverage their bargaining power to squeeze the upstream enterprise's profit margins and limit their growth potential. This increases the business risks for firms and erodes their profitability through the customers' "predatory effect", ultimately affecting the firms' ability to invest in R&D [3]. Meng et al. [43] found through empirical results that customer concentration hinders firms' technological innovation only when customers' bargaining power is strong. Furthermore, the industrial organization analysis argues that increased customer integration weakens a firm's market control, while successful innovation requires strong market control [19]. As customer integration weakens a firm's market control, firms often reduce R&D investment to avoid risk due to their limited risk tolerance. Zhao et al. [1] through the data of listed companies in the manufacturing industry found that customer concentration will continue to inhibit the innovation output of the enterprise, over-concentration of customer relationships to increase the business risk, enterprises to reduce the risk of investment to reduce the investment of innovation resources, which in turn has an impact on the innovation performance of the enterprise. In the absence of external R&D funding, it becomes difficult for firms to maintain previous levels of innovation efficiency, which further affects their overall innovation performance. High customer concentration often results in a lack of product diversity, compelling firms to cater to the production demands of key customers. This reliance on established purchasing habits stifles innovation and limits the expansion of sales channels. Additionally, constrained by tight budgets and risk

aversion, firms may choose to reduce innovation investments. In the event of a sudden interruption of the contract between the customer and the enterprise, the enterprise will not only face the risk of stagnant sales, but also lack the ability to adapt to the modern market innovation because of over-reliance on the products demanded by the old customers. Based on the above analysis, the following hypothesis is proposed.

H2 Customer integration has a negative correlation with firms' innovation performance.

## Supplier integration and firms' innovation performance

Suppliers, as crucial stakeholders, play a key role in the upstream of the supply chain. In an increasingly competitive market, key suppliers often determine critical factors such as cost, quality and delivery time, which indirectly affects the manufacturer's production capabilities. Therefore, firms need to continuously adapt their supply chain strategies to keep pace with rapidly changing market conditions and consumer preferences [44]. From an innovation process perspective, supplier involvement enhances the acquiring of innovation resources and optimizes the R&D process [45]. The involvement of suppliers in early production research and development with their own empirical skills and expertise can reduce the huge workload caused by innovation uncertainty and avoid the impact of unknown factors on the ability to control new products. From an innovation outcome perspective, Estrada et al. [46] found that sharing information and resources between business partners to develop inter-organizational synergies mobilizes effective resources for precise input and enhances innovative output, thereby improving enterprise innovation performance and the ability to generate innovation revenues.

However, the impact of supplier integration on innovation performance is not always positive. Based on Porter's theory of industrial positioning, there are five fundamental forces that manipulate the competitive situation in an industry, and supplier bargaining power is one of them [47]. Supplier's position and bargaining power increases as the firm's dependence on the supplier increases, and the purchase price of raw materials and the quality of products are affected. Sun et al. [48] found in his research that in the information transfer law of trade network, the high bargaining power party has more willingness to transfer information technology, that is, the direction of information technology transfer is from the high bargaining power party to the low bargaining power party. Therefore, when the relative bargaining power of enterprises is weaker, their obstacles to the acquisition of information and technology become larger, which in turn affects the output of innovation performance. On the other hand, in today's competitive and volatile market environment, where the firms in the supply chain maintain the traditional mindset of competing for benefits, the behavior of supplier integration, as a risk factor, tends to undermine the benefits of cooperation that have been developed over time between the two partners. When the supply sources of enterprises are concentrated in a few major suppliers, the autonomy and flexibility of enterprises' decision-making are weakened, which makes the enterprises' ability to resist risks weak, and thus weakens their willingness to innovate and research and development [49]. Moreover, as supplier integration increases, so does the firm's path dependence. Long-term cooperation with a few suppliers can lead to a lock-in effect, risking knowledge leakage and high costs when switching suppliers. Ju et al. [50] found that communication, cooperation and integration between firms and suppliers have a significant impact on the development of supply chain resilience. Fixed supplier relationships may exacerbate disrupted supply chain relationships, while having alternate suppliers contributes to supply chain resilience [51]. Organizations inherently possess inertia, which arises from the inevitable complexity of interdependent structures, routines, and roles [52]. With the addition of path dependence and an unstable supply chain environment, it is difficult for firms to increase the investment in R&D funding,

and corporate innovation is even more hindered. Based on the above analysis, the following hypotheses are proposed.

H3 Supplier integration has a negative correlation with firms' innovation performance.

## Supply chain integration, R&D investment intensity, and firms' innovation performance

R&D investment plays a crucial role in fostering innovation within manufacturing companies. R&D investment facilitates the creation of new products, processes, designs, and technologies, and enhances existing products. It leads to improved innovation performance, often measured by patents [53]. Various levels of R&D investment yield different impacts on innovation performance [27]. Firms allocate R&D funds based on their unique traits and the current market environment to maximize profitability [17]. According to the theory of resource dependence, although manufacturing enterprises can learn new technologies and knowledge through cooperation with suppliers and customers, the new things they receive also need a certain amount of R&D funding to be fully absorbed and transformed into products and results. This suggests that a reasonable allocation of R&D funds can enhance a company's ability to absorb and convert innovations when engaging in customer integration and supplier integration. Additionally, during collaboration with key customers and suppliers, both upstream and downstream partners focus on the enterprise's level of R&D investment. Focusing resources on R&D strengthens relationships and business confidence between firms and their partners, fostering information exchange and closer communication, ultimately enhancing innovation performance. From the empirical level, Wang et al. [54] found that a reasonable and effective allocation of technological innovation resources directly affects the results of technological innovation and promotes the rapid conversion of technological innovation results, the acceleration of the R&D process and the reduction of development costs so that the formation of intangible creativity and control of the enterprise, the enterprise can significantly enhance the level of technological innovation performance. Xie et al. [55] found that companies with higher R&D investment intensity benefited more from cooperation. As a result, R&D investment intensity contributes to the development of an enterprise's ability to absorb information, enabling it to maximize value from its upstream and downstream partners while managing its relationships with key suppliers and key customers. R&D investment intensity directly influences the R&D process and the efficient use of external collaborative resources. Reasonable allocation of R&D funds not only speeds up product development within enterprises, but also helps to strengthen the effective sharing of internal and external resources, which positively affects the integrated supply chain management. Based on the above analysis, the following hypotheses are proposed:

H4 R&D investment intensity positively moderates the relationship between customer integration and firm innovation performance.

H5 R&D investment intensity positively moderates the relationship between supplier integration and firm innovation performance.

## Research design

### Data sources

Panel data of A-share listed manufacturing companies in China's Shanghai and Shenzhen markets from 2012–2021 are selected, and the classification of manufacturing industries is based on the Guidelines for Industry Classification of Listed Companies, which was revised

by the Securities and Futures Commission in 2012. First, samples with significant missing or abnormal data were excluded, as well as listed companies categorized as ST or ST*. These companies have experienced consecutive losses for two or three years, and their management models have undergone significant changes due to shifts in the business environment, which could distort the empirical results if included. Second, as customer and supplier integration indicators are tied to a company's operational strategy and not all firms disclose this data, samples with missing key variables were excluded, and any missing data across consecutive years were filled using interpolation. Finally, to minimize the impact of some abnormal values on the overall data, continuous variables were shrink-tailed by 1% and 99% before and after. Based on the above data screening methods, 9847 panels of data from 1038 firms were finally collected. The internal control index for the manufacturing industry was sourced from the DIB database, while other data were obtained from the CSMAR database. All the data are primary data.

## Variable selection

**Dependent variable. Innovation performance (Patent):** Past scholars have found that different indicators overlap with each other by examining a variety of corporate innovation performance measures, so multiple indicators to measure innovation performance are likely to have an impact on the empirical results. The number of patent applications, as the total number of patent applications filed by enterprises in the same year, can directly reflect the activity of technological activities and the enthusiasm of enterprises to seek patent protection, and this indicator has the advantages of universality, originality and consistency [56]. Therefore, this paper draws on the studies of Jiang et al. [56] and Ji et al. [57], this paper selects the combined number of patent applications (Patent) for design patents, utility model patents, and invention patents as an indicator to measure the innovation performance of enterprises.

**Independent variables. Internal integration (LNICI):** The internal of manufacturing firms emphasizes cross-functional collaboration and synergy between different departments, and its response is more similar to that of the internal control index response. Qi et al. [58] believe that the status of internal control of domestic listed companies can be reflected by the internal control index, which is widely used by scholars and has scientific validity. Therefore, this paper draws on this scholar's approach and chooses the internal control index (LNICI) as a measure of internal integration in manufacturing companies.

**Customer integration (CI):** Customer integration emphasizes the coordinated operation of the firm with its customers through practices, behavioral control, etc., and also reflects the extent of the firm's dependence on its customers. Therefore, this paper refers to the research method of Liang et al. [59], and this paper chooses the top five customers' annual total sales share (CI) to indicate the degree of customer integration.

**Supplier integration (SI):** Supplier integration emphasizes the frequency of cooperation between the enterprise and its core suppliers, and ensures the efficient operation of materials and the coordination of organizational processes according to the behavioral habits of the two parties in their transactions. The number of business transactions and the number of product transactions between an enterprise and its top five suppliers can reflect the close relationship between the top five suppliers and the enterprise. Therefore, this paper draws on Zeng [60] and selects the top five suppliers' share of total annual business (SI) to indicate the degree of supplier integration.

**Moderator variable. Research and development investment intensity (R&D):** R&D investment as the enterprise technology innovation activities of financial support and material protection can be represented by the absolute amount of enterprise R&D expenditure, but

due to the level of China's manufacturing industry gap degree is large, different kinds of manufacturing industry between the amount of input expenditure lack of comparability, the corresponding variables should be able to adapt to the enterprise scale and enterprise market position [61]. In this paper, we refer to the indicator selection of Ali et al. [62] and Lv et al. [63] and use the (RD) of the ratio of R&D investment to operating income as the indicator of measuring R&D investment in this paper.

**Control variables.** Referring to the studies of Meng et al. [43], Jiang [56], Su et al. [64], this paper controls the variables of the firm's gearing ratio (Lev), formula age of establishment (FirmAge), cash flows (cashflow), return on total assets (ROA), growth rate of operating income (Growth), and Tobin's Q value (TobinQ). Table 1 summarizes these variables.

## Model construction

In this study, we refer to the work of Huang et al. [65] to construct a model that examines the potential impact of internal integration, customer integration, and supplier integration on corporate innovation performance. The selection of this model is based on several key factors: First, it fully considers the potential impact of temporal shocks on corporate innovation performance by incorporating year fixed effects, aiming to enhance the accuracy and reliability of the regression results. Second, the inclusion of control variables in the model helps to eliminate the interference of other potential factors, ensuring an accurate assessment of the impacts of different dimensions of supply chain integration. Lastly, multiple regression analysis is employed to clearly reveal the direct relationships under different dimensions, facilitating a comprehensive understanding of the results related to supply chain integration and corporate innovation performance.

$$Patent = \beta_0 + \beta_1 LNICI + \beta_2 Lev + \beta_3 ROA + \beta_4 FirmAge + \beta_5 Growth + \beta_6 Cashflow + \beta_7\ TobinQ + \sum Year + \varepsilon \tag{1}$$

$$Patent = \beta_0 + \beta_1 CI + \beta_2 Lev + \beta_3 ROA + \beta_4 FirmAge + \beta_5 Growth + \beta_6 Cashflow + \beta_7\ TobinQ + \sum Year + \varepsilon \tag{2}$$

$$Patent = \beta_0 + \beta_1 SI + \beta_2 Lev + \beta_3 ROA + \beta_4 FirmAge + \beta_5 Growth + \beta_6 Cashflow + \beta_7\ TobinQ + \sum Year + \varepsilon \tag{3}$$

**Table 1. Variable definitions.**

| Variable | Symbol | Definition |
| --- | --- | --- |
| Innovation performance | Patent | Total number of patent applications filed by enterprises in the year |
| Internal control index | LNICI | Internal control index disclosed by Dibble Data Information |
| Customer integration | CI | Percentage of total annual sales to top five customers |
| Supplier integration | SI | Percentage of total annual business of top five suppliers |
| Research and development investment intensity | RD | R&D investment/revenue |
| Corporate debt ratio | Lev | Total liabilities/total assets |
| Net profit margin on total assets | ROA | Net profit/average annual total assets |
| Years of incorporation | FirmAge | ln(Current year minus year of incorporation + 1) |
| Tobin's Q value | TobinQ | (Market value of outstanding shares + market value of preferred shares + liabilities)/Total assets |
| Revenue growth rate | Growth | Current year's operating income/previous year's operating income −1 |
| Cash flow | Cashflow | Net operating cash flow/total assets |

In Eqs (1), (2), and (3), patent denotes innovation performance, LNICI denotes internal integration, CI denotes customer integration, SI denotes supplier integration, the rest of the variables are control variables, and *Year* denotes the annual dummy variable. $\beta_0$ is the intercept, and $\varepsilon$ denotes the random error term.

## Empirical analysis

### Descriptive statistics

This paper analyzes the main variables with descriptive statistics. As shown in Table 2, the minimum value of Patent is 0 and the maximum value is 727, and the minimum value of LNICI is 0 and the maximum value is 831.9, which is well represented because of the large sample capacity and the large span of the value domain. The average value of Patent is 35.32, which indicates that the overall level of corporate innovation ability of the listed companies in the manufacturing industry of the A-share companies in Shanghai and Shenzhen is not high. Further to do the test on the multicollinearity of the variables, it is found that the variance expansion factors of the variables in the model are all less than 2, which is much smaller than the critical value of 10, proving that the variables in this paper do not have the problem of multicollinearity.

### Regression analysis

Table 3 presents the benchmark regression results of the effects of internal integration, customer integration, and supplier integration on corporate innovation performance in columns (1, 2), and (3), respectively. Column (1) shows the impact of internal integration on corporate innovation performance. The coefficient of LNICI is 0.019, which is significant at the 1% level. This indicates a positive relationship between internal integration and corporate innovation performance. Internal integration helps optimize the allocation of internal resources, enhances knowledge and information sharing among departments, reduces internal friction and resource waste, and thereby boosts the overall innovation capacity of the firm. Hence, Hypothesis H1 is supported. Column (2) displays the results of the impact of customer integration on corporate innovation performance. The coefficient of CI is −0.114, significant at the 5% level. This reflects a negative relationship between customer integration and corporate innovation performance. High levels of customer integration require significant resource investment to meet customers' personalized demands, which can detract from the resources available for innovation. Moreover,

**Table 2. Descriptive statistics.**

| VarName | Obs | Mean | SD | Min | Max |
|---|---|---|---|---|---|
| Patent | 9,847 | 35.32 | 96.35 | 0 | 727 |
| LNICI | 9,847 | 649.3 | 101.9 | 0 | 831.9 |
| SI | 9,847 | 32.40 | 18.35 | 1.770 | 89.49 |
| CI | 9,847 | 28.70 | 19.07 | 3.120 | 87.50 |
| RD | 9,847 | 4.418 | 3.650 | 0 | 21.80 |
| Lev | 9,847 | 0.395 | 0.185 | 00563 | 0.803 |
| ROA | 9,847 | 0.0444 | 0.0561 | −0.152 | 0.208 |
| FirmAge | 9,847 | 2.899 | 0.315 | 1.946 | 3.466 |
| Growth | 9,847 | 0.145 | 0.297 | −0.422 | 1.621 |
| Cashflow | 9,847 | 0.0541 | 0.0615 | −0.1119 | 0.227 |
| TobinQ | 9,847 | 2.0986 | 1.2193 | 0.8736 | 7.4271 |

**Table 3. Test results.**

|  | (1) | (2) | (3) | (4) | (5) | (6) |
|---|---|---|---|---|---|---|
|  | Patent | Patent | Patent | Patent | Patent | Patent |
| LNICI | 0.019*** |  |  | 0.021*** |  |  |
|  | (3.32) |  |  | (2.67) |  |  |
| CI |  | −0.114** |  |  | −0.151* |  |
|  |  | (−2.25) |  |  | (−1.70) |  |
| SI |  |  | −0.131*** |  |  | −0.208** |
|  |  |  | (−2.94) |  |  | (−2.42) |
| _cons | 9.699 | 28.497 | 38.160** | −19.484*** | 1.051 | 4.552 |
|  | (0.51) | (1.51) | (1.99) | (−3.43) | (0.13) | (0.55) |
| Control | YES | YES | YES | YES | YES | YES |
| Year_FE | YES | YES | YES | YES | YES | YES |
| Obs | 9847 | 9847 | 9847 | 9847 | 9847 | 9847 |
| r2_a | 0.0306 | 0.0303 | 0.0304 | 0.0370 | 0.0437 | 0.0453 |

*, **, And *** passed the significance test at the level of 10%, 5%, and 1%, respectively.

excessive customer integration can reduce the channels through which customer demands are reflected, thereby lowering the accessibility to customer needs and inhibiting innovation output. Thus, Hypothesis H2 is confirmed. Column (3) presents the test results for the impact of supplier integration on corporate innovation performance. The coefficient of SI is −0.131, also significant at the 1% level. This indicates that supplier integration negatively impacts corporate innovation performance. Increased supplier integration can limit the firm's access to heterogeneous resources, which are crucial for driving R&D investment and innovation. Consequently, higher levels of supplier integration reduce the motivation for innovation investment, thereby affecting corporate innovation performance. Thus, Hypothesis H3 is verified.

## Robustness test

In order to enhance the robustness of the regression results, this study conducts a series of robustness tests. First, the measure of firms' innovation performance is changed. Referring to the study of Chen et al. [66], the logarithm of the number of invention patent applications plus one is used as a proxy variable for firm innovation. The regression results using this measure are consistent with the previous paper, confirming the robustness of the main conclusions of this paper.

Second, considering the possible time lag in the impact of supply chain integration on firms' innovation performance, this study regresses the measures of internal integration, customer integration, and supplier integration all one period lagged. The results obtained are also consistent with the above regression results.

Thirdly, referring to the practice of Huang et al. [65], a new round of empirical tests is conducted by applying the panel-corrected standard error model. The regression results are shown by columns 4, 5, and 6 of Table 3, and the results of each test are consistent with the conclusions of the previous analysis.

## Mechanism analysis

In order to assess the mechanisms by which supply chain integration affects firms' innovations, and given the apparent flaws in causal inference in the three-stage mediating mechanism test, this study follows the approach of Wang et al. [54]. We constructed the interaction

model described in Eqs (4) and (5). Considering that the impact of customer integration and supplier integration on firms' innovation performance may be affected by R&D investment, this paper introduces R&D investment intensity as a moderating variable to examine its moderating effect on the relationship between external integration and firms' innovation performance. If R&D investment does help external integration to improve firms' innovative performance, the coefficient of the interaction term should be significantly positive.

$$Patent = \beta_0 + \beta_1 CI + \beta_2 CI * RD + \beta_3 Lev + \beta_4 ROA + \beta_5 FirmAge + \beta_6 Growth \\ + \beta_7 Cashflow + \beta_8 TobinQ + \sum Year + \varepsilon \tag{4}$$

$$Patent = \beta_0 + \beta_1 SI + \beta_2 SI * RD + \beta_3 Lev + \beta_4 ROA + \beta_5 FirmAge + \beta_6 Growth \\ + \beta_7 Cashflow + \beta_8 TobinQ + \sum Year + \varepsilon \tag{5}$$

First, as shown in column 1 of the Table 4, customer integration and supplier integration still significantly inhibit innovation performance at the 1% level, while the coefficients of the respective interaction terms of customer integration and supplier integration with R&D investment intensity are significantly positive, which suggests that customer integration and supplier integration are still detrimental to firms' innovation performance, but R&D investment intensity will inhibit the detrimental effects of customer integration and supplier integration on firms' innovation performance, and hypotheses H4 and H5 have been verified. Based on the increase of supplier integration, the bargaining power of the suppliers will produce the occupying behavior of the enterprise's capital, and neither the heterogeneous resources outside the enterprise nor the turnover cycle and abundance of the internal capital will be sufficient to support the enterprise's innovation activities. At the same time, the higher the degree of customer integration, the more customers need more business behavior protection, customers will delay payment, the enterprise's own accounts receivable and notes receivable increase, which has a direct impact on the enterprise's cash flow. At this time, R&D investment helps the enterprise to solve the cash

**Table 4. Mechanism test.**

|  | (1) | (2) |
|---|---|---|
|  | Patent | Patent |
| CI | −0.171*** |  |
|  | (−2.86) |  |
| SI |  | −0.188*** |
|  |  | (−3.61) |
| RD × CI | 0.013* |  |
|  | (1.77) |  |
| RD × SI |  | 0.015** |
|  |  | (2.09) |
| _cons | 25.311 | 25.962 |
|  | (1.35) | (1.38) |
| Control | YES | YES |
| Year_FE | YES | YES |
| Obs | 9847 | 9847 |
| r2_a | 0.0302 | 0.0305 |

*, **, And *** passed the significance test at the level of 10%, 5%, and 1%, respectively.

[https://doi.org/10.1371/journal.pone.0316251.t004](https://doi.org/10.1371/journal.pone.0316251.t004)

flow problem, and at the same time reduces the risk of the enterprise's capital being occupied, positively moderating the relationship between supplier cohesion, customer integration and the enterprise's innovation performance. When enterprises exchange resources with suppliers and customers in the open market system, the intensity of R&D investment guarantees the maximization of the output of resource transactions, thus moderating the negative relationship between external integration and enterprise innovation performance.

## Heterogeneity test

**The effect of the nature of the business on outcomes.** Compared with non-state-owned enterprises, state-owned enterprises can have more resources for market competition due to their political connections and social capital. They face fewer difficulties in obtaining advantageous resources and enjoy greater policy support, making their innovation output less reliant on supply chain integration compared to non-state-owned enterprises. Thus, this paper expects that the effect of supply chain integration on corporate innovation is more significant in non-SOEs. The samples are grouped according to the nature of enterprises, with SOEs coded as 1 and non-SOEs coded as 0. The regression results are shown in Table 5, with columns (1, 2) and (3) representing the regression results for the sample of state-owned enterprises, and columns (4, 5) and (6) representing the regression results for the sample of non-state-owned enterprises. It is found that the internal integration of SOEs plays a more significant role in influencing firms' innovation performance compared to non-SOEs. While in non-state-owned enterprises, the impact of supplier integration and customer integration on enterprise innovation performance is more significant. As an important carrier of the national economy, the upstream and downstream suppliers and customers of state-owned enterprises are fully screened through strict comparative evaluation, and the most suitable suppliers and customers are finally selected to cooperate with them. The impact of external integration on innovation performance is mitigated through effective relationship and risk management, ensuring quality information exchange and stable cooperation.

Further analysis reveals that the moderating role of R&D investment intensity on the relationship between external integration and firms' innovation performance varies by firm type. Columns (1) and (2) of Table 6 present the results for state-owned enterprises, while columns

Table 5. The heterogeneity analysis.

|  | (1) | (2) | (3) | (4) | (5) | (6) |
|---|---|---|---|---|---|---|
|  | Patent | Patent | Patent | Patent | Patent | Patent |
| LNICI | 0.062*** |  |  | 0.010** |  |  |
|  | (3.03) |  |  | (2.18) |  |  |
| CI |  | −0.206 |  |  | −0.098** |  |
|  |  | (−1.47) |  |  | (−2.34) |  |
| SI |  |  | −0.206 |  |  | −0.124*** |
|  |  |  | (−1.60) |  |  | (−3.45) |
| _cons | −197.209** | −151.058* | −150.812* | 27.734** | 38.706*** | 39.993*** |
|  | (−2.41) | (−1.84) | (−1.84) | (2.15) | (3.05) | (3.16) |
| Control | YES | YES | YES | YES | YES | YES |
| Year_FE | YES | YES | YES | YES | YES | YES |
| Obs | 3164 | 3164 | 3164 | 6683 | 6683 | 6683 |
| r2_a | 0.0281 | 0.0266 | 0.0270 | 0.0386 | 0.0393 | 0.0390 |

*, **, And *** passed the significance test at the level of 10%, 5%, and 1%, respectively.

**Table 6. The heterogeneity analysis.**

|  | (1) | (2) | (3) | (4) |
|---|---|---|---|---|
|  | Patent | Patent | Patent | Patent |
| CI | −0.301* |  | −0.152*** |  |
|  | (−1.85) |  | (−3.05) |  |
| RD×CI | 0.031 |  | 0.011** |  |
|  | (1.12) |  | (2.01) |  |
| SI |  | −0.289** |  | −0.182*** |
|  |  | (−2.00) |  | (−4.26) |
| RD×SI |  | 0.036 |  | 0.012** |
|  |  | (1.26) |  | (2.46) |
| _cons | −159.128* | −159.689* | 36.375*** | 37.184*** |
|  | (−1.94) | (−1.95) | (2.89) | (2.96) |
| Control | YES | YES | YES | YES |
| Year_FE | YES | YES | YES | YES |
| Obs | 3164 | 3164 | 6683 | 6683 |
| r2_a | 0.0263 | 0.0270 | 0.0391 | 0.0391 |

*, **, And *** passed the significance test at the level of 10%, 5%, and 1%, respectively.

(3) and (4) present the results for non-state-owned enterprises. The results show that the moderating effect of R&D investment intensity on external integration and firm innovation performance is more pronounced in non-state-owned firms. This is because, unlike state-owned firms that have access to higher quality heterogeneous resources, non-state-owned firms must rely on fast and accurate market judgment and efficient enterprise management to ensure sustained competitiveness of their new products. Therefore, compared with state-owned enterprises, the moderating influence of R&D investment intensity of non-state-owned enterprises is better. Secondly, compared with non-state-owned enterprises, state-owned enterprises can obtain more policy support, and when faced with changes in market demand, due to the weaker sense of market foresight, resulting in a lack of incentive to take R&D investment. Additionally, unlike non-SOEs, the top management of SOEs is influenced by administrative pressures and promotion incentives. In the face of intense market competition, SOEs are more likely to adopt conservative administrative tactics rather than higher-risk R&D investment strategies.

**The effect of equity concentration on outcomes.** Equity concentration reflects the influence of large shareholders on the business strategy decisions, and different equity concentration can produce different governance effects [67]. Some studies suggest that the shareholders in enterprises with higher equity concentration exert stronger over the company [68]. This stronger control reduces the level of diversification and affects the enterprise's diversification strategy. Other studies argue that equity concentration is a quantitative measure of shareholder concentration, reflected in the proportion of company shares held by shareholders. It directly represents the degree of influence on the board of directors in formulating innovative strategic decisions [69]. To explore the relationship between supply chain integration and corporate innovation performance more deeply, this paper uses the shareholding proportion of the top five shareholders as an indicator of equity concentration. Columns (1, 2) and (3) of Table 7 represent the sample group of enterprises with high equity concentration, while columns (4, 5) and (6) represent the sample group of enterprises with low equity concentration. The results show that the effect of internal integration

**Table 7. The heterogeneity analysis.**

|  | (1) | (2) | (3) | (4) | (5) | (6) |
|---|---|---|---|---|---|---|
|  | Patent | Patent | Patent | Patent | Patent | Patent |
| LNICI | 0.040*** |  |  | 0.016** |  |  |
|  | (3.91) |  |  | (2.03) |  |  |
| CI |  | −0.051 |  |  | −0.187** |  |
|  |  | (−0.76) |  |  | (−2.44) |  |
| SI |  |  | −0.141** |  |  | −0.204*** |
|  |  |  | (−2.35) |  |  | (−2.94) |
| _cons | −23.884 | 5.315 | 9.331 | −4.340 | 13.510 | 14.272 |
|  | (−1.15) | (0.27) | (0.47) | (−0.15) | (0.48) | (0.51) |
| Control | YES | YES | YES | YES | YES | YES |
| Year_FE | YES | YES | YES | YES | YES | YES |
| Obs | 4809 | 4809 | 4809 | 5038 | 5038 | 5038 |
| r2_a | 0.0427 | 0.0408 | 0.0411 | 0.0228 | 0.0239 | 0.0232 |

*, **, And *** passed the significance test at the level of 10%, 5%, and 1%, respectively.

on innovation performance is more significant in firms with high equity concentration compared to those with low equity concentration. In contrast, the negative effects of customer and supplier integration on innovation performance are more pronounced in firms with low equity concentration. This is because majority shareholders in firms with high equity concentration are more motivated and willing to supervise and intervene in management's supply chain decisions due to their higher shareholding ratio. This reduces opportunistic behavior by management and mitigates the negative impact of external integration on innovation performance, known as the "supervisory effect" of equity concentration [70].

Further exploration reveals that equity concentration also moderates the relationship between external integration and firms' innovation performance. Columns (1) and (2) of Table 8 represent the sample of high equity concentration firms and columns (3) and (4) represent the sample of low equity concentration firms. The results show that the moderating effect of R&D investment intensity between external integration on firms' innovation performance is more significant for firms with low equity concentration relative to firms with high equity concentration. Majority shareholders often fear the inherent risks of innovation and may choose to halt R&D projects to protect their own interests. Therefore, under the same level of R&D investment intensity, majority shareholders worry that changes in the enterprise's organization and business model may threaten their position. They may leverage their control and resource dominance, enabled by their high shareholding ratio, to create a "hollowing out effect" on the company, thereby reducing innovation performance.

## Discussion

This paper primarily investigates the impact of different forms of supply chain integration on corporate innovation performance from the perspective of supply chain structure. In the context of innovation-driven economic development, the discussion is undoubtedly of both theoretical and practical significance. Our findings show that internal integration within manufacturing firms not only facilitates more effective sharing of market and operational information but also enables a more accurate understanding of market demand changes. This enhances the timeliness, precision, and consistency of operational decisions across departments, ensuring the circulation of key innovation elements within the firm and thereby

**Table 8.  The heterogeneity analysis.**

|  | (1) | (2) | (3) | (4) |
|---|---|---|---|---|
|  | Patent | Patent | Patent | Patent |
| CI | −0.118 |  | −0.310*** |  |
|  | (−1.45) |  | (−3.48) |  |
| RD × CI | 0.015 |  | 0.029*** |  |
|  | (1.45) |  | (2.71) |  |
| SI |  | −0.186*** |  | −0.312*** |
|  |  | (−2.70) |  | (−3.88) |
| RD × SI |  | 0.013 |  | 0.026*** |
|  |  | (1.29) |  | (2.59) |
| _cons | 2.809 | 6.632 | 6.450 | 8.093 |
|  | (0.14) | (0.33) | (0.23) | (0.29) |
| Control | YES | YES | YES | YES |
| Year_FE | YES | YES | YES | YES |
| Obs | 4809 | 4809 | 5038 | 5038 |
| r2_a | 0.0401 | 0.0407 | 0.0239 | 0.0237 |

*, **, And *** passed the significance test at the level of 10%, 5%, and 1%, respectively.

improving corporate innovation performance. Regarding external integration, both customer and supplier integration can pose risks, influenced by the business environment and management strategies, which continuously affect innovation output. Resource dependence theory can explain why some key enterprises within the supply chain choose not to increase reliance on external strategic partners to save costs but rather invest in retaining critical resources internally to create a unique competitive advantage [71]. Moreover, as firms increase their upstream and downstream partnerships, resulting in reduced levels of customer and supplier integration, the diversity and richness of collaborative relationships grow. This broader network grants firms access to a wider range of external innovation resources, fostering complementary resource advantages with partners and consequently enhancing innovation performance.

Additionally, the study finds that the intensity of R&D investment positively moderates the relationship between customer integration, supplier integration, and corporate innovation performance. In other words, R&D investment effectively mitigates the adverse impacts of customer and supplier integration on innovation performance. Customer and supplier integration influence the firm's heterogeneous resources, and increased R&D investment enhances the ability to withstand potential risks, improves sensitivity to market trends, and aids in optimizing resource allocation [72]. R&D investment enables firms to more effectively absorb and utilize feedback and new technologies from customers and suppliers, transforming them into innovation opportunities and tangible innovation outcomes.

Further research reveals that the impact of supply chain integration on corporate innovation performance exhibits significant heterogeneity across different corporate contexts. In state-owned enterprises and firms with high equity concentration, the positive impact of internal integration on innovation performance is particularly pronounced. In SOEs, abundant resources are often dispersed across various departments or subsidiaries. Internal integration can effectively consolidate these resources, enhancing internal control and supervision, reducing communication and coordination costs between departments, and

thereby improving resource utilization efficiency and innovation performance. High equity concentration typically signifies centralized decision-making, enabling swift coordination and allocation of internal resources, avoiding resource wastage or redundant construction, and promoting optimal resource allocation through internal integration to foster innovation. Conversely, non-state-owned enterprises and firms with low equity concentration often rely on market and customer feedback to guide their innovation direction. Due to the lack of strong internal decision-making authority, these firms may overly depend on customers' current demands and preferences, overlooking potential market changes and long-term technological innovation. This short-sightedness leads to incremental innovation rather than breakthrough innovation, ultimately limiting the firm's competitiveness in rapidly changing markets. Additionally, firms that heavily rely on specific suppliers tend to exhibit inertia in technology choices and innovation pathways, making disruptive innovation difficult. Supplier integration tends to solidify supply chain and production processes, increasing resistance to innovation and reducing the motivation for innovation. In non-SOEs and firms with low equity concentration, customer and supplier integration involves substantial communication and coordination efforts, which increase operational complexity and costs. Significant resources devoted to coordinating and integrating with customers and suppliers may reduce the resources available for internal R&D and innovation. Therefore, R&D investment is more effective in mitigating the negative impact of customer and supplier integration on innovation performance in these firms.

This study makes contributions in three main areas. First, in terms of research content, it helps to enrich the theoretical framework of corporate innovation performance. By analyzing the impact of integration across three dimensions of the supply chain on a firm's innovation performance, it provides new empirical evidence on how supply chain integration affects innovation in manufacturing firms. While some studies have discussed the relationship between supply chain integration and innovation, the results remain contentious. This research empirically examines various aspects of this relationship, thereby contributing to the existing literature in this field. Second, regarding the research mechanism, this study constructs a theoretical framework that incorporates R&D investment, external integration, and corporate innovation performance. This enhances our understanding of the role of R&D investment in corporate innovation. Additionally, based on resource dependence theory, it offers new mitigation pathways for studying supply chain integration, presenting theoretical value. This new perspective can aid further analyses of how supply chain integration impacts corporate innovation performance. Finally, this study provides decision-making support for manufacturing firms across regions aiming for high-quality innovation in the current competitive market environment. By exploring how differences in corporate nature or equity concentration affect innovation from a supply chain management perspective, it finds that the three dimensions of supply chain integration impact innovation differently based on the nature of the firm and the level of equity concentration. Moreover, the moderating effect of R&D investment varies accordingly. This study contributes to existing research on the moderating role of supply chains in corporate innovation and provides valuable insights for firms seeking to enhance their innovation capabilities.

## Conclusion

### Research findings

In the era of a rapidly advancing knowledge economy driven by technological progress, traditional management models have become increasingly passive and slow to respond to market changes. Consequently, companies are actively seeking to build a comprehensive competitive

advantage through supply chain integration. Against this macro background, this paper selects data from A-share listed manufacturing companies in China as the research sample, taking into account the characteristics of manufacturing firms in supply chain relationship management. The study delves into the impact of supply chain integration on corporate innovation performance. The findings reveal that internal supply chain integration has a significantly positive effect on corporate innovation performance, while customer integration and supplier integration are negatively correlated with corporate innovation performance. Robustness tests confirm the reliability of these results. Mechanism analysis indicates that the moderating effect of R&D investment intensity helps mitigate the negative impact of customer integration and supplier integration on corporate innovation performance. Heterogeneity analysis shows that the impact of supply chain integration on corporate innovation performance varies across firms with different ownership types and equity concentration levels, and the moderating role of R&D investment intensity is also influenced by these factors. In state-owned enterprises and firms with high equity concentration, the positive impact of internal integration on corporate innovation performance is more pronounced. In non-state-owned enterprises and firms with low equity concentration, the negative impact of customer integration and supplier integration on corporate innovation performance is more significant, and R&D investment intensity is more effective in mitigating the negative impact of customer and supplier integration on innovation performance.

## Managerial implications

This study provides valuable practical insights for business management. The managerial implications of our research include the following: First, manufacturing firms must emphasize internal environmental governance. Leveraging advanced digital technologies can reduce information and organizational barriers, enabling regular cross-departmental communication through information-sharing platforms. When integrating internal business units, firms should avoid a uniform resource integration model. Instead, they should adjust resource allocation levels based on business attributes, current priorities, and short-term or long-term benefit categories, gradually forming an adaptable integration system. This approach minimizes internal resource competition, concentrates resources efficiently, reduces management costs, and leverages economies of scale for innovative output. Second, when integrating the upstream and downstream segments of the supply chain, manufacturing firms should deeply engage in supply chain practices and strive to build collaborative innovation relationships from a comprehensive supply chain perspective. They should balance the power distribution among market participants by fully utilizing economies of scale and scope. It is essential to shift away from traditional business models dominated by a few key suppliers and customers. Instead, firms should dynamically adjust supply chain members based on the operational status and functional positioning of each node enterprise within the supply chain. Additionally, supply chain partners need to establish physical platforms for R&D resource investment and collaborate with venture capital and innovation service organizations to facilitate resource sharing and technological exchange, supporting joint innovation. Lastly, different types of enterprises should adopt different supply chain cooperation management methods. In selecting transactional contracts, firms should avoid relational transactions and use rigorous contracts and breach costs to constrain the economic behavior of supply chain enterprises during cooperation, thereby mitigating opportunism and transaction risks due to information asymmetry. In regions with underdeveloped property rights protection, the stability and effectiveness of relational transactions are often poor, making it challenging to manage and allocate innovative outputs within the supply chain. Therefore, it is necessary to establish and improve long-term mechanisms for supply

chain management and the division of innovation outputs, and to dynamically adjust these mechanisms based on environmental conditions in future collaborations.

## Research limitations

While this study draws significant conclusions through empirical analysis, the reliability of the empirical results may be limited by the sample data's source and diversity. Constrained by the data released by listed companies, the study faces limitations in the choice of variables for measuring supply chain integration. Additionally, supply chain relationships exist in non-listed companies across various regions. Future research should include a wider range of enterprises to enhance the generalizability of the findings. Secondly, this study finds that customer integration and supplier integration inhibit corporate innovation. However, it does not specifically identify the threshold levels of integration, which could provide more detailed guidance for businesses. Future research could select several typical manufacturing companies of different types and employ methods such as individual case studies and in-depth interviews to explore manufacturing innovation projects in depth. This would help reveal how manufacturing firms integrate their supply chains to drive innovation in dynamic environments.

## Author contributions

**Conceptualization:** Juanmei Zhou, Jie Mei.

**Data curation:** Jie Mei.

**Formal analysis:** Juanmei Zhou, Jie Mei.

**Funding acquisition:** Juanmei Zhou.

**Investigation:** Jie Mei.

**Methodology:** Jie Mei.

**Project administration:** Juanmei Zhou.

**Resources:** Juanmei Zhou, Jie Mei.

**Software:** Jie Mei.

**Validation:** Jie Mei.

**Visualization:** Jie Mei.

**Writing – original draft:** Jie Mei.

**Writing – review & editing:** Juanmei Zhou, Jie Mei.

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
