## [Decision Letter · Decision Letter 0]

27 May 2024

PONE-D-24-06764Research on the Impact of Supply Chain Integration on the Innovation Performance of Manufacturing Enterprises --Based on the Moderating Role of R&D Investment IntensityPLOS ONE

Dear Dr. Zhou,

Thank you for submitting your manuscript to PLOS ONE. After careful consideration, we feel that it has merit but does not fully meet PLOS ONE’s publication criteria as it currently stands. Therefore, we invite you to submit a revised version of the manuscript that addresses the points raised during the review process.

**ACADEMIC EDITOR: **

Dear Author

My own reading of your paper placed me largely in agreement with the reviewers, who is generally positive but also raised a number of important questions and concerns about the current draft of your paper. You need to focus on the following areas:

1. The gap should be clearly mentioned.

2. Results and discussion should be improved and explained in detail and discussed with latest literature.

Overall, it is clear that there is promise in your paper, yet at present the paper is not ready for publication. Please understand that this decision does not guarantee

eventual publication in the journal.

If you decide to revise and resubmit your paper, please include an itemized cover letter in which you respond in detail to ALL comments by the reviewers and mine. Please highlight changes in manuscript draft as well.

We look forward to receiving your revised manuscript.

Kind regards,

Afshan Naseem, Ph.D.

Academic Editor

PLOS ONE

Journal Requirements:

3. Thank you for stating the following financial disclosure: "Funding: Humanities and Social Sciences Foundation of the Ministry of Education (22YJC790142); Humanities and Social Sciences Foundation of the Ministry of Education (202104031402090); Teaching Reform and Innovation Project of Higher Education Institutions in Shanxi Province (J2021343)" 

Additional Editor Comments:

Dear Author

My own reading of your paper placed me largely in agreement with the reviewers, who is generally positive but also raised a number of important questions and concerns about the current draft of your paper. You need to focus on the following areas:

1. The gap should be clearly mentioned.

2. Results and discussion should be improved and explained in detail and discussed with latest literature.

3. The paper should be proof read.

Overall, it is clear that there is promise in your paper, yet at present the paper is not ready for publication. Please understand that this decision does not guarantee

eventual publication in the journal.

If you decide to revise and resubmit your paper, please include an itemized cover letter in which you respond in detail to ALL comments by the reviewers and mine. Please highlight changes in manuscript draft as well.

Reviewers' comments:

Reviewer's Responses to Questions

**Comments to the Author**

1. Is the manuscript technically sound, and do the data support the conclusions?

Reviewer #1: Partly

Reviewer #2: Partly

2. Has the statistical analysis been performed appropriately and rigorously? 

Reviewer #1: Yes

Reviewer #2: Yes

3. Have the authors made all data underlying the findings in their manuscript fully available?

Reviewer #1: Yes

Reviewer #2: No

4. Is the manuscript presented in an intelligible fashion and written in standard English?

Reviewer #1: Yes

Reviewer #2: Yes

5. Review Comments to the Author

Reviewer #1: Here's a breakdown of how to revise your paper for PLOS ONE publication based on the reviewer's comments:

Abstract:

Clearly State Aims and Research Gap: Rewrite the abstract to explicitly state the research objectives, the identified gap in current knowledge, and how your study aims to address it. Include brief mentions of your methodology and key findings.

Literature Review:

Highlight Significance and Update References: Strengthen your literature review by emphasizing the importance of your research within the existing body of knowledge. Include 3-5 recent references (published in 2024) from PLOS ONE articles that directly address your research topic or methodology. For each reference, incorporate relevant extracts that support your study and showcase the significance of your contribution. Use the PLOS ONE reference style guide for formatting.

Methodology:

Explain and Motivate Research Objectives and Methodology: Provide a more detailed explanation of your research objectives and methodology. Motivate your choices by explaining why these specific methods were selected and how they address your research questions.

Deepen Description of Mathematical Model and Variables: Expand on the description of your mathematical model and the variables used. Define them clearly and explain their role in the analysis.

Results and Discussion:

Sample Size Information: Include details about your data collection process, particularly the sample size, data collection period, and the rationale for choosing these specific parameters.

Supporting Evidence for Conclusions: Address any assumptions made in your analysis. Explain how your conclusions are supported by your theoretical framework, empirical data, or literature review findings.

Discussion:

Deeper Discussion of Implications: Expand on the potential impact of your study. Discuss how decisions across various fields can benefit from the insights gained from your research.

Conclusion:

Develop a Dedicated Conclusion Section: Create a separate conclusions section that summarizes your key findings, reiterates your unique contributions (both theoretical and practical), acknowledges limitations of the research, and proposes directions for future studies.

Final Touches:

Proofread Thoroughly: Perform a meticulous final proofread to ensure clarity, grammar, and formatting accuracy.

By following these steps and addressing the reviewer's specific concerns, you can significantly improve your paper's chances of successful publication in PLOS ONE.

Reviewer #2: Reviewer’s comments

Topic: Research on the Impact of Supply Chain Integration on the Innovation Performance of Manufacturing Enterprises --Based on the Moderating Role of R&D Investment Intensity

Suggestion: I suggest the authors should be given an opportunity to improve the study based on my comments provided.

Introduction

The paper empirically examines the impact of supply chain integration on firms' innovation performance through panel data of listed manufacturing companies, and provides an in-depth discussion of the relationship between the two based on the perspective of R&D investment intensity.

General comments: The study is relevant because it can added supply chain knowledge especially with evident from developing economy. The study has some potential value but can improved by adhering to the following suggestions.

Topic and introduction

1. I suggest you trimmed the topic to a more precise form. You can remove the “research on…”, “based on,,,” portions. Please refer to other studies in this journal properly shape the topic.

2. Also avoid using abbreviations in the topic.

3. The authors must use more current or recent articles, and properly pitch the gap in extant literature, to bring out the novelty of the study.

4. The gap is not clearly stated. The authors must clearly explain the problem found in extant literature. The gap for both the direct and the moderating relationships must be established. It appears the background information has buried the problem the study aimed to address. So please streamline the background information properly.

5. Also, the authors must cite authoritative statements. For example, in the introduction, the authors stated that “At present, China's economic development model is changing from a crude to an innovation-driven model” but this statement was without any citation. Why? Such deficiencies must be corrected throughout the manuscript.

6. Please convert your gaps into research questions. You can refer to: Aryee, R., Alfa, A. A., Acquah, H., Addey, G. B., & Akoto, E. J. K. (2024). Circular economy, customer citizenship behaviour and firm performance: Some empirical evidence. Business Strategy & Development, 7(2), e377. https://doi.org/10.1002/bsd2.377

Theoretical background

1. The section 2 should first capture and describe the theory used in the study. (its concept, variables or assumptions and applicability in the domain). Then it was used to distill or generate the various hypotheses. More importantly, insights from the theory employ (together with literature) should be used to develop the hypotheses. This is somehow lacking in this manuscript and must be corrected.

2. Make sure your in-text citation style (throughout the whole manuscript) is consistent with the standards of the journal

Methodology

1. What is the research design? This was not stated.

2. Did you use primary data or secondary or both? This must be clear in the work.

3. The study’s methodology lacks some key components. The methodology should be properly structured to include research design, research approach, study area, sources and types of data, population, sampling procedure, instruments, data collection procedures, operationalization of variables, data analysis and ethical issues.

4. Concerning the data analysis, it appears the study used the multiple regression analysis technique. Why was this done? Especially when the model has several variables. Such situations (complex models) calls for more robust multivariate analytical technique such as structural equation modeling (SEM). For this is the reason why SEM was developed (see Hair Jr. et al., 2017)….to handle complex models. If possible, please use SEM either the variance or covariance based in your work. Anyway, if you still think the multiple regression is okay, then please provide amble justification for itsusage (instead of SEM) in the manuscript.

5. The results (on beta coefficient, p-values, R2 etc.) can be presented in a more acceptable manner, See Aryee et al., (2024). Follow the logical representation this paper.

5. I cannot find the analysis or results of the interaction effects….you can graph the moderating effect to bring more clarity, as was done in Aryee et al., (2024).

Conclusion and discussion

1. Why do you capture conclusion and discussion together. Why not rather have results and discussion together. Then under conclusion you can have sub-sections on implication, limitations and future research directions.

2. Finally, proofread the entire manuscript properly.

6. PLOS authors have the option to publish the peer review history of their article (what does this mean? ). If published, this will include your full peer review and any attached files.

**Do you want your identity to be public for this peer review?** For information about this choice, including consent withdrawal, please see our Privacy Policy .

Reviewer #1: No

Reviewer #2: No

---

## [Author Response · Author response to Decision Letter 0]

3 Jul 2024

Thank you very much for your suggestions on this article, according to your tips article to make the following changes.

1. I suggest you trimmed the topic to a more precise form. You can remove the “research on…”, “based on,,,” portions. Please refer to other studies in this journal properly shape the topic.

Thank you to the reviewers for their suggestions.

This paper has modified its own title by removing parts of it based on other research in journals.

2. Also avoid using abbreviations in the topic.

Thank you to the reviewers for their suggestions.

Based on the reviewer's comments, avoid abbreviations in the title and write R&D inputs as full names.

3. The authors must use more current or recent articles, and properly pitch the gap in extant literature, to bring out the novelty of the study.

Thank you to the reviewers for their suggestions.

More than 20 newer and recent articles have been added to the original, and the introductory section also describes how this study fills a gap in the existing literature, thus highlighting the novelty of the study.

4. The gap is not clearly stated. The authors must clearly explain the problem found in extant literature. The gap for both the direct and the moderating relationships must be established. It appears the background information has buried the problem the study aimed to address. So please streamline the background information properly.

Thank you to the reviewers for their suggestions.

The third and fourth paragraphs of the introduction have been revised in the light of the lack of clarity on the gaps, and the gaps in direct and regulating relationships have been additionally identified through more informative literature and scholarly perspectives. Some of the background information has also been streamlined as requested.

5. Also, the authors must cite authoritative statements. For example, in the introduction, the authors stated that “At present, China's economic development model is changing from a crude to an innovation-driven model” but this statement was without any citation. Why? Such deficiencies must be corrected throughout the manuscript.

Thank you to the reviewers for their suggestions.

Relevant authoritative statements are included in the introduction, drawing out the context of the article through citations. Missing citations have been added.

6. Please convert your gaps into research questions. You can refer to: Aryee, R., Alfa, A. A., Acquah, H., Addey, G. B., & Akoto, E. J. K. (2024). Circular economy, customer citizenship behaviour and firm performance: Some empirical evidence. Business Strategy & Development, 7(2), e377. https://doi.org/10.1002/bsd2.377

Thank you to the reviewers for their suggestions.

This article by Aryee has been referenced as per the reviewer's comments, in terms of the content and the entire structure of the article.

7. The section 2 should first capture and describe the theory used in the study. (its concept, variables or assumptions and applicability in the domain). Then it was used to distill or generate the various hypotheses. More importantly, insights from the theory employ (together with literature) should be used to develop the hypotheses. This is somehow lacking in this manuscript and must be corrected.

Thank you to the reviewers for their suggestions.

Already based on the reviewer's comments, the theory used in the study is firstly described in the second part to generate various hypotheses through grounded theory. And have utilized elements from the theories used to formulate the content of my own hypotheses. This has been revised in both articles.

8. Make sure your in-text citation style (throughout the whole manuscript) is consistent with the standards of the journal.

Thank you to the reviewers for their suggestions.

This paper has modified its own citation style based on the journal's template and related articles so that it meets the journal's requirements.

9. What is the research design? This was not stated.

Thank you to the reviewers for their suggestions.

The research design section has been supplemented to deepen this part of the argument.

10. Did you use primary data or secondary or both? This must be clear in the work.

Thank you to the reviewers for their suggestions.

All the data used in this article are primary data, which through modifications have been reflected in the in the article.

11. The study’s methodology lacks some key components. The methodology should be properly structured to include research design, research approach, study area, sources and types of data, population, sampling procedure, instruments, data collection procedures, operationalization of variables, data analysis and ethical issues.

Thank you to the reviewers for their suggestions.

Based on the issues raised by the reviewers, the research methodology section was revised to ensure that this section was rationalized. It was ensured that the research methodology of the paper included the research design, research methodology, sources and types of data, use of instruments, data collection process, treatment of variables and analysis of data.

12. Concerning the data analysis, it appears the study used the multiple regression analysis technique. Why was this done? Especially when the model has several variables. Such situations (complex models) calls for more robust multivariate analytical technique such as structural equation modeling (SEM). For this is the reason why SEM was developed (see Hair Jr. et al., 2017)….to handle complex models. If possible, please use SEM either the variance or covariance based in your work. Anyway, if you still think the multiple regression is okay, then please provide amble justification for its usage (instead of SEM) in the manuscript.

Thank you to the reviewers for their suggestions.

We thank the reviewers for their valuable comments on our manuscript. We understand the advantages of structural equation modeling in dealing with complex models. However, in this study, we chose to use multiple regression analysis based on the following considerations. First, based on the research objectives and hypotheses of this paper, our study aims to test specific causal relationships and direct effects between variables. Multiple regression analysis can clearly reveal these direct relationships and the results are easy to interpret and understand. Secondly, based on practicality and interpretability, the results of multiple regression analysis are more intuitive and easy to interpret, especially for readers and practical application scenarios. We hope that our findings can provide a direct reference for policy makers and practitioners, and the results of multiple regression are more advantageous in this regard. Finally, based on the reading of previous related literature, the research on supply chain integration is mostly developed based on multiple regression modeling, in addition to the questionnaire approach, the empirical approach. This indicates that it is an effective and commonly used approach in our research context. For example, XU X.M,QUAN X.F,ZHU S.S. Supply Chain Concentration and Corporate Innovation: An Empirical Study Based on Listed Companies in China's Manufacturing Industry [J]. Business Economics and Management, 2022, (04): 5-16. DOI:10.14134/j.cnki.cn33-1336/f.2022.04.001; Su T.Y, Yu Y.Z. A study on the impact of supply chain integration on corporate environmental performance in Chinese manufacturing companies[J]. Social Science Journal,2023(06):183-190; HUANG J.G,JI X.X,LI Y.X. The impact of corporate innovation strategy on business performance - A case study of supply chain integration strategy in manufacturing companies[J]. Scientific Management Research,2021,39(01):111-115.DOI:10.19445/j.cnki.15-1103/g3.2021.01.018.

13. The results (on beta coefficient, p-values, R2 etc.) can be presented in a more acceptable manner, See Aryee et al., (2024). Follow the logical representation this paper.

Thank you to the reviewers for their suggestions.

The coefficients and p-values etc. at the conclusion of this paper have been modified with reference to Aryee et al. and the style of statements in other papers in this journal.

14. I cannot find the analysis or results of the interaction effects….you can graph the moderating effect to bring more clarity, as was done in Aryee et al., (2024).

Thank you to the reviewers for their suggestions.

Based on the format of other articles in the journal, this study placed the moderating effects in the later section on mechanism testing. The description of the relevant interaction effect analysis and results was comprehensively improved, and the description of this section was exacerbated to facilitate readers' ability to understand clearly.

15. Why do you capture conclusion and discussion together. Why not rather have results and discussion together. Then under conclusion you can have sub-sections on implication, limitations and future research directions.

Thank you to the reviewers for their suggestions.

Reference has been made to other papers in this journal to revise the conclusion section. Discussion and conclusion sections have been added, while relevant content such as management comments and research limitations have been refined to meet journal requirements.

16. Finally, proofread the entire manuscript properly.

Thank you to the reviewers for their suggestions.

Has correctly proofread the entire manuscript in accordance with the relevant requirements of the journal.

17. Clearly State Aims and Research Gap: Rewrite the abstract to explicitly state the research objectives, the identified gap in current knowledge, and how your study aims to address it. Include brief mentions of your methodology and key findings.

Thank you to the reviewers for their suggestions.

The summary has been revised in its entirety. First, the purpose of the study has been clearly stated in accordance with the journal requirements. Also the research gap content has been added subsequently. The research methodology and key findings are also mentioned in the abstract.

18. Highlight Significance and Update References: Strengthen your literature review by emphasizing the importance of your research within the existing body of knowledge. Include 3-5 recent references (published in 2024) from PLOS ONE articles that directly address your research topic or methodology. For each reference, incorporate relevant extracts that support your study and showcase the significance of your contribution. Use the PLOS ONE reference style guide for formatting.

Thank you to the reviewers for their suggestions.

The literature review section of this paper has been revised to emphasize the importance and update the references as requested by the reviewers. Also, 3-4 recent references published in PLOS ONE in 2024 have been added to the revised section to emphasize the importance of the study and to update the references. For these new references, a theoretical analysis section has been added to help improve this study.

19. Explain and Motivate Research Objectives and Methodology: Provide a more detailed explanation of your research objectives and methodology. Motivate your choices by explaining why these specific methods were selected and how they address your research questions.

Thank you to the reviewers for their suggestions.

The research methodology and other elements of the methodology have been revised based on the comments of the reviewers. Elements such as research design and research methodology have been expanded to clarify how the specific methodology chosen for this study addresses the research questions in this paper.

20. Deepen Description of Mathematical Model and Variables: Expand on the description of your mathematical model and the variables used. Define them clearly and explain their role in the analysis.

Thank you to the reviewers for their suggestions.

This paper expands on the construction of the mathematical model and the description of the variables. Descriptions of the mathematical model and the variables used have been added so that it is clear what role these elements play in this paper.

21. Sample Size Information: Include details about your data collection process, particularly the sample size, data collection period, and the rationale for choosing these specific parameters.

Thank you to the reviewers for their suggestions.

The process of data collection, and sample size etc. has been described deeper in the article. The rationale for choosing the parameters of these variables has also been deepened discussed.

22. Supporting Evidence for Conclusions: Address any assumptions made in your analysis. Explain how your conclusions are supported by your theoretical framework, empirical data, or literature review findings.

Thank you to the reviewers for their suggestions.

Relevant theories and supporting evidence have been included in the Getting Results section. Thus, it better explains how its own conclusions are supported.

23. Deeper Discussion of Implications: Expand on the potential impact of your study. Discuss how decisions across various fields can benefit from the insights gained from your research.

Thank you to the reviewers for their suggestions.

An in-depth discussion of the possible impact of this paper is added at the conclusion. Several perspectives are discussed on how various fields could benefit from this study.

24. Develop a Dedicated Conclusion Section: Create a separate conclusions section that summarizes your key findings, reiterates your unique contributions (both theoretical and practical), acknowledges limitations of the research, and proposes directions for future studies.

Thank you to the reviewers for their suggestions.

A separate conclusion section has been created in response to reviewer comments, summarizing its own key findings and highlighting the unique contributions of this paper. The limitations of the study are also addressed and lead to possible future research directions in this area and how the limitations encountered in this paper can be improved.

25. Proofread Thoroughly: Perform a meticulous final proofread to ensure clarity, grammar, and formatting accuracy.

Thank you to the reviewers for their suggestions.

The entire article has been meticulously proofread to ensure that it meets the grammatical and formatting requirements of this journal. And the language has been touched up throughout the text.

By following these steps and addressing the reviewer's specific concerns, you can significantly improve your paper's chances of successful publication in PLOS ONE.

---

## [Decision Letter · Decision Letter 1]

1 Sep 2024

PONE-D-24-06764R1Supply chain integration and innovation performance of manufacturing firms: The moderating role of research and development investment intensityPLOS ONE

Dear Dr. Zhou,

Thank you for submitting your manuscript to PLOS ONE. After careful consideration, we feel that it has merit but does not fully meet PLOS ONE’s publication criteria as it currently stands. Therefore, we invite you to submit a revised version of the manuscript that addresses the points raised during the review process.

**ACADEMIC EDITOR: ** Dear Author

I agree with comments of reviewers. Although, there is a potential in the manuscript but if it gets improved as follows:

You should improve references.

The manuscript should be proof read by a language expert to remove grammatical mistakes.

We look forward to receiving your revised manuscript.

Kind regards,

Afshan Naseem, Ph.D.

Academic Editor

PLOS ONE

Journal Requirements:

Additional Editor Comments:

Dear Author

I agree with comments of reviewers. Although, there is a potential in the manuscript but if it gets improved as follows:

You should improve references.

The manuscript should be proof read by a language expert to remove grammatical mistakes.

Reviewers' comments:

Reviewer's Responses to Questions

**Comments to the Author**

1. If the authors have adequately addressed your comments raised in a previous round of review and you feel that this manuscript is now acceptable for publication, you may indicate that here to bypass the “Comments to the Author” section, enter your conflict of interest statement in the “Confidential to Editor” section, and submit your "Accept" recommendation.

Reviewer #3: (No Response)

Reviewer #4: All comments have been addressed

Reviewer #5: All comments have been addressed

2. Is the manuscript technically sound, and do the data support the conclusions?

Reviewer #3: Yes

Reviewer #4: Yes

Reviewer #5: Yes

3. Has the statistical analysis been performed appropriately and rigorously? 

Reviewer #3: Yes

Reviewer #4: Yes

Reviewer #5: Yes

4. Have the authors made all data underlying the findings in their manuscript fully available?

Reviewer #3: Yes

Reviewer #4: Yes

Reviewer #5: Yes

5. Is the manuscript presented in an intelligible fashion and written in standard English?

Reviewer #3: No

Reviewer #4: Yes

Reviewer #5: Yes

6. Review Comments to the Author

Reviewer #3: The research topic is interesting. However, the citations are not up-to-date and based on Chinese Journal. In addition, the basic punctuation usage is incorrect, Grammar is not good, and the reference writing has the mistakes. Please revise it before recommending the paper in details. I'd be delighted to review the revised version of the paper.

Reviewer #4: (No Response)

Reviewer #5: - The manuscript appears technically sound based on the methodology and data analysis conducted. The use of multiple regression models, control variables, and robustness checks suggests that the experiments were conducted rigorously with appropriate replication.

- The manuscript is generally written in standard English and is intelligible, with clear organization and logical flow between sections. However, minor grammatical errors and typographical issues were noted, which should be corrected to improve the overall readability.

7. PLOS authors have the option to publish the peer review history of their article (what does this mean? ). If published, this will include your full peer review and any attached files.

**Do you want your identity to be public for this peer review?** For information about this choice, including consent withdrawal, please see our Privacy Policy .

Reviewer #3: **Yes: ** Asst.Prof.Dr.Wissawa Aunyawong

Reviewer #4: **Yes: ** Madeleine Delmore

Reviewer #5: No

---

## [Author Response · Author response to Decision Letter 1]

20 Sep 2024

Thank you very much for your suggestions on this article, according to your tips article to make the following changes.

1. I agree with comments of reviewers. Although, there is a potential in the manuscript but if it gets improved as follows:

You should improve references.

The manuscript should be proof read by a language expert to remove grammatical mistakes.

Thank you to the reviewers for their suggestions.

Changes have been made to the article's references as requested by the academic editor. And the grammatical errors in the article have been scrutinized and revised to ensure better quality content.

Thank you to the reviewers for their suggestions.

The reference list of the article has been revised according to the requirements of the journal. Duplicate references are eliminated and the format of the references is modified according to the journal format.

The citation [3] in the references was deleted, so the citation order of the whole film was adjusted. Refer to "Can co-ordination advancement of two-way FDI improve resource misallocation? Evidence from 285," published July 8 in the journal Plos One Format requirements for cities in China and journals. The original quoted content is directly written as the Statistical Bulletin of China's science and technology Investment in 2022,...

The reference citation [43] was deleted. It was found that it was cited twice, and the order of subsequent articles was adjusted.

The cited literature [30] has been modified. Since the citations in the original part are quite old, we have searched for the latest theoretical content and updated this part to the newer literature. The content of the article has also made relevant changes, Change the original text to "Theoretically, resource dependence theory accommodates the simultaneous exploration of the firm's internal environment and external influences, not only elaborating on the importance of internal resources, but also emphasizing the important driving force of the firm's decision-making when subject to external constraints. "is changed to: “Theoretically, resource dependence theory accommodates the simultaneous exploration of the firm's internal environment and external influences, not only explaining the importance of internal organizational coordination, but also emphasizing the interdependent collaboration between firms and their partners.”

At the same time, some older Chinese periodicals have been modified and replaced with newer foreign periodicals. Change citation [52] to [50] Mastio EA, Clegg SR, Pina e Cunha M, Dovey K. Leadership ignoring paradox to maintain inertial order. Journal of Change Management. 2024; 24 (2) : 83-101.

Change citation [63] to [61] Ali S, Muhammad H, Migliori S. R&D investment and SMEs performance: the role of capital structure decisions. EuroMed Journal of Business. 2024.

3. The research topic is interesting. However, the citations are not up-to-date and based on Chinese Journal. In addition, the basic punctuation usage is incorrect, Grammar is not good, and the reference writing has the mistakes. Please revise it before recommending the paper in details. I'd be delighted to review the revised version of the paper.

Thank you to the reviewers for their suggestions.

According to the opinions of reviewers, several old Chinese periodicals are updated to the latest foreign periodicals. The original citation [52] was changed from a Chinese journal to a foreign journal. And according to the new quoted point of view to change the content of the article, beautify the content of the article on the basis of also updated the quoted. the document changes the original "There is inertia in the organization itself" to "Organizations inherently possess inertia, which arises from the inevitable complexity of interdependent structures, routines, and roles.

The older Chinese articles citing [63] are also changed to the latest foreign journals. In terms of the measurement of R&D investment indicators, I checked the latest foreign journals and found that the measurement methods were all the same, so I replaced them with the latest foreign literatures.

Secondly, I carefully checked the grammar and punctuation in the article, and corrected the wrong parts. The sentence of the article is also polished to some extent. The format of the references is also modified using plos one format in endnote software.

4. The manuscript appears technically sound based on the methodology and data analysis conducted. The use of multiple regression models, control variables, and robustness checks suggests that the experiments were conducted rigorously with appropriate replication.

The manuscript is generally written in standard English and is intelligible, with clear organization and logical flow between sections. However, minor grammatical errors and typographical issues were noted, which should be corrected to improve the overall readability.

Thank you to the reviewers for their suggestions.

The article has been carefully checked and revised, some grammatical errors and typographical problems have been solved, and the language of the article has been polished to some extent.

---

## [Decision Letter · Decision Letter 2]

9 Dec 2024

Supply chain integration and innovation performance of manufacturing firms: The moderating role of research and development investment intensity

PONE-D-24-06764R2

Dear Dr. Mei,

We’re pleased to inform you that your manuscript has been judged scientifically suitable for publication and will be formally accepted for publication once it meets all outstanding technical requirements.

Kind regards,

Afshan Naseem, Ph.D.

Academic Editor

PLOS ONE

Additional Editor Comments (optional):

Reviewers' comments:

Reviewer's Responses to Questions

**Comments to the Author**

1. If the authors have adequately addressed your comments raised in a previous round of review and you feel that this manuscript is now acceptable for publication, you may indicate that here to bypass the “Comments to the Author” section, enter your conflict of interest statement in the “Confidential to Editor” section, and submit your "Accept" recommendation.

Reviewer #3: All comments have been addressed

Reviewer #5: (No Response)

2. Is the manuscript technically sound, and do the data support the conclusions?

Reviewer #3: Yes

Reviewer #5: (No Response)

3. Has the statistical analysis been performed appropriately and rigorously? 

Reviewer #3: Yes

Reviewer #5: (No Response)

4. Have the authors made all data underlying the findings in their manuscript fully available?

Reviewer #3: Yes

Reviewer #5: (No Response)

5. Is the manuscript presented in an intelligible fashion and written in standard English?

Reviewer #3: Yes

Reviewer #5: (No Response)

6. Review Comments to the Author

Reviewer #3: According to the revised version of the article, the grammatical errors and typographical issues are revised. Moreover, for the issue that the citations are not up-to-date and based on Chinese Journal is revised. However, please add more citations from Asian scholars to ensure the various references worldwide. The following citations are suggested:

Aunyawong, W., Wararatchai, P., & Hotrawaisaya, C. (2018). The mediating role of trust among supply chain partners on supply chain integration, cultural intelligence, logistics flexibility and supply chain performance. Science International Journal, 30(4), 629-633.

Aunyawong, W., Wararatchai, P., & Hotrawaisaya, C. (2020). The influence of supply chain integration on supply chain performance of auto-parts manufacturers in Thailand: a mediation approach. International Journal of Supply Chain Management, 9(3), 578-590.

You can add the above citations to the following texts because such citations discuss the customer integration that enhances supply chain managment opearion by which the first paper is a concept paper indexed in Web of Science and the second paper is the reseach aricle indexed in SCOPUS.

"Customers, being directly in front of consumer groups, are responsible for ensuring consumers

are aware of and purchase the company's products. At the same time, enterprises can obtain

feedback from customers on product design, usage performance, and other market elements,

which not only enhances the visibility of external problems, but also helps enterprises gain a

clear insight into market changes and needs, thus ensuring that enterprises can make better and

faster decisions on supply chain management operations ***[3x, 3x]***"

Reviewer #5: (No Response)

7. PLOS authors have the option to publish the peer review history of their article (what does this mean? ). If published, this will include your full peer review and any attached files.

**Do you want your identity to be public for this peer review?** For information about this choice, including consent withdrawal, please see our Privacy Policy .

Reviewer #3: No

Reviewer #5: No

---

## [Editor Report · Acceptance letter]

PONE-D-24-06764R2

PLOS ONE

Dear Dr. Mei,

I'm pleased to inform you that your manuscript has been deemed suitable for publication in PLOS ONE. Congratulations! Your manuscript is now being handed over to our production team.

Kind regards,

on behalf of

Dr. Afshan Naseem

Academic Editor

PLOS ONE